# The Applications and Potentials of Extracellular Vesicles from Different Cell Sources in Periodontal Regeneration

**DOI:** 10.3390/ijms24065790

**Published:** 2023-03-17

**Authors:** Xin Huang, Huiyi Wang, Chuan Wang, Zhengguo Cao

**Affiliations:** 1The State Key Laboratory Breeding Base of Basic Science of Stomatology (Hubei-MOST) & Key Laboratory of Oral Biomedicine Ministry of Education, School & Hospital of Stomatology, Wuhan University, Wuhan 430079, China; 2Department of Periodontology, School & Hospital of Stomatology, Wuhan University, Wuhan 430079, China

**Keywords:** bacteria, extracellular vesicle, mammal, periodontal regeneration, periodontitis, plant

## Abstract

Periodontitis is a chronic infectious disease worldwide that can cause damage to periodontal supporting tissues including gingiva, bone, cementum and periodontal ligament (PDL). The principle for the treatment of periodontitis is to control the inflammatory process. Achieving structural and functional regeneration of periodontal tissues is also essential and remains a major challenge. Though many technologies, products, and ingredients were applied in periodontal regeneration, most of the strategies have limited outcomes. Extracellular vesicles (EVs) are membranous particles with a lipid structure secreted by cells, containing a large number of biomolecules for the communication between cells. Numerous studies have demonstrated the beneficial effects of stem cell-derived EVs (SCEVs) and immune cell-derived EVs (ICEVs) on periodontal regeneration, which may be an alternative strategy for cell-based periodontal regeneration. The production of EVs is highly conserved among humans, bacteria and plants. In addition to eukaryocyte-derived EVs (CEVs), a growing body of literature suggests that bacterial/plant-derived EVs (BEVs/PEVs) also play an important role in periodontal homeostasis and regeneration. The purpose of this review is to introduce and summarize the potential therapeutic values of BEVs, CEVs and PEVs in periodontal regeneration, and discuss the current challenges and prospects for EV-based periodontal regeneration.

## 1. Introduction

Periodontitis is a chronic inflammatory disease characterized by irreversible destruction of periodontal supporting tissues, which is related to the disturbance of the local microbial community, and then immune cell infiltration, osteoclast activation and inflammatory cytokine release induced by the overactivation of host immunity. It is also associated with obesity, cardiovascular disease, Alzheimer’s disease and many other systemic diseases [1,2]. Further, periodontal disease was reported as a risk factor for infection after mandibular fracture and associated with low cognitive performance amongst older adults [3,4]. The traditional treatment for periodontitis focuses on removing calculus, plaque biofilm and diseased cementum through scaling and root planning to effectively control the inflammation. However, it is still faced with great difficulties in the reconstruction of damaged periodontal tissues [5].

Periodontal regeneration includes the regeneration of alveolar bone, cementum, PDL and gingiva. Among them, bone regeneration is the most studied and the most promising. In contrast, orderly and dense PDL regeneration and cementum regeneration on root surface are more challenging [6]. Cementum is a thin layer of mineralized tissue similar to bone tissue located on the surface of tooth root, which is very important for maintaining the stability and health of the tooth. Hence, cementum regeneration was considered to be the golden standard for evaluating a successful whole periodontal regeneration [7,8]. PDL connects alveolar bone and cementum by thick collagen bundles. PDL is a highly vascularized tissue between the bone and cementum, containing a variety of cells such as fibroblasts, endothelial cells, epithelial cells, nerve cells, and undifferentiated mesenchymal cells. Periodontal ligament cells (PDLCs) are able to differentiate into osteoblasts and cementoblasts under appropriate conditions, and are essential for maintaining the homeostasis of periodontal tissues [9].

Research in periodontal regeneration included guided tissue regeneration, platelet-rich plasma and fibrin, growth and differentiation factors, enamel matrix protein, and autologous graft, allograft and xenografts [10,11,12,13]. Nevertheless, periodontal tissue is a system containing a variety of cells and complex structures. Most of the above approaches can only exert a limited and unstable effect in specific circumstances, and need further clinical verification [14,15]. To date, cell therapy based on stem cells and immune cells has been widely concerned in periodontal tissue regeneration. A range of stem cells such as bone marrow mesenchymal stem cells (BMMSCs), periodontal ligament stem cells (PDLSCs), DPSCs (dental pulp stem cells), and a series of immune cells such as neutrophils and macrophages have been proved to play an important role in periodontal regeneration [16,17,18,19,20]. However, they both have some limitations, such as dedifferentiation during stem cell expansion, reduced regeneration efficiency after transplantation, unstable phenotypes of immune cells after transplantation, effects of the environment in different patients on immune cell phenotypes, and inconsistent standards for the production, transportation, storage, and safety of transplanted cells [16,21,22].

EVs are bilayer membranous particles secreted by cells into the extracellular space, which were originally thought to be a pathway of cellular waste excretion [23]. In addition to being the carrier of waste, it also contains a large number of bioactive materials (proteins, nucleic acids, lipids, metabolites) for cell communications [24]. The biological functions of EVs are largely determined by their source cells. More and more studies have shown that EVs derived from stem/immune cells (SCEVs/ICEVs) are the main paracrine participants account for the periodontal regeneration [19,25,26]. Compared with cell therapy, EVs has many advantages including lower immune clearance rate, long storage time, non-toxicity (avoid DMSO and other toxic substances mixed), delivery of active substances more efficiently, strong penetrability through narrow spaces such as blood–brain barrier [21,27]. So, EVs is emerging as an alternative to cell therapy for tissue regeneration [21,28].

In recent years, EV-based periodontal regeneration has become a research front. The generation of EVs is highly conserved between humans, bacteria, and plants. In addition to CEVs, a large number of studies have revealed that BEVs and PEVs are also associated with periodontal disease, homeostasis maintenance and periodontal regeneration [29,30]. There are similarities and differences between them, and some research on them is growing. Therefore, the aim of this review is to sort out and discuss the potential therapeutic values of the above three vesicles in periodontal regeneration. Further, the current challenges and prospects for EV-based periodontal regeneration were also summarized.

## 2. CEVs and Periodontal Regeneration

### 2.1. Classification and Biogenesis of CEVs

Although these vesicles have a similar structure and physical properties, they are heterogeneous in size, contents, biological function, and cell type of origin. Therefore, their naming is still controversial [27,31]. According to the classification recommendations from the International Society for Extracellular Vesicles (ISEV), CEVs can be classified into small EVs (<200 nm) and medium/large EVs (>200 nm) on the basis of their size [32]. EVs can be derived from different cell culture supernatants or body fluids. Based on the current published literature, CEVs were divided into exosomes (also known as small EVs, sEVs; <200 nm), microvesicles (50–1000 nm) and apoptotic bodies (50–5000 nm) [31].

The biogenesis of CEVs varies, mainly including the following aspects. (1) The process of exosomes formation consists of early sorting endosomes (ESEs) formed by the first plasma membrane invagination; late sorting endosomes (LSEs) formed by ESEs maturation; LSEs form multivesicular bodies (MVBs) containing multiple intraluminal vesicles (ILVs) by the second membrane invagination; MVBs can fuse with autophagosomes and lysosomes for degradation, or fuse with the plasma membrane to release ILVs (or exosomes) into the extracellular environment. MVBs can be formed through endosomal sorting complex required for transport (ESCRT)-dependent and ESCRT-independent pathways, which determines that exosomes have a variety of protein markers such as CD9, CD63, TSG101, Alix, CD81, HSP90 [31,33]. (2) Microvesicles were formed through the direct outward budding of the plasma membrane. This process requires cytoskeleton contraction, phospholipid redistribution, and Ca^2+^-dependent enzymatic mechanism. The typical inclusions include CD40, selectin, integrin, cytoskeleton protein, and cholesterol. ARF6 and VCAMP3 were also identified as protein markers [34,35,36]. (3) Different from the above two, apoptotic bodies were produced by programmed dying cells which undergo nuclear chromatin contraction, plasma membrane budding, and cell segmentation. Similar to microvesicles, biogenesis of apoptotic bodies also depends on actin-myosin interaction. The phosphatidylserine on their surface binding with Annexin V can be recognized, engulfed, and degraded by macrophages soon after release. The larger apoptotic bodies have more abundant cell-derived components, and phosphatidylserine, genomic DNA, Annexin V, platelet reactive protein and C3b were considered as the biomarkers of apoptotic bodies [37,38,39,40].

CEVs from multiple cells, especially SCEVs and ICEVs, play a positive role in various tissue regeneration including periodontal regeneration [21,28]. In this part, we summarized the effects of SCEVs and ICEVs on different periodontal cell functions and periodontal regeneration, as well as their potential mechanisms.

### 2.2. CEVs Regulate Periodontal Immunity and Inflammation

Inflammation is the response of the immune system to harmful stimuli, such as pathogens, damaged cells, toxic compounds or radiation. Periodontitis can be caused by a variety of periodontal pathogens. Controlling inflammation is a prerequisite for periodontal regeneration [1,41]. CEVs are closely related to inflammation. In terms of pathogenesis, CEVs can act as a shuttle or biomarker for inflammation, and were involved in the progression of various tissue inflammation [42]. In terms of therapy, a large number of studies have shown that SCEVs and ICEVs (sEVs were mainly discussed below) have powerful immunomodulatory and anti-inflammatory effects on various inflammatory diseases such as periodontitis [43,44,45].

There are many studies on SCEVs regulating immune cell functions and alleviating periodontal inflammation. Related research reported that stem cells from human exfoliated deciduous teeth (SHED)-derived CEVs (SHED-EVs) reduced *IL-6* and *TNF-α* expression in lipopolysaccharide (LPS)-induced inflammatory BMMSCs, and in tissues from the defect area of periodontitis mice [46]. SCEVs could also regulate the cytokine expression in macrophages. For example, PDLC/PDLSC-derived CEVs (PDLC/PDLSC-EVs) could inhibited cell pyroptosis and the production of inflammatory cytokines (IL-1β, IL-18 and TNF-α) in macrophages stimulated with LPS through NF-κB signaling pathway or delivering *miR-590-3p* silencing TLR4 and its downstream signaling pathway, to modulate the periodontal microenvironment [47,48]. CEVs derived from BMMSCs (BMMSC-EVs) was demonstrated to inhibit the inflammatory response of macrophages induced by *Porphyromonas gingivalis* (*P. gingivalis*), the keystone pathogen of periodontitis, and reduce the cell infiltration in rat periodontitis tissues [49]. Compared with BMMSC-EVs, gingival mesenchymal stem cell (GMSC)-derived CEVs (GMSC-EVs) promoted the expression of IL-10, Arg-1, CD206 and inhibit the expression of TNF-α in LPS-treated monocytes more effectively. GMSC-EVs could also influence T cell functions by suppressing CD4^+^ T cell activation, promoting Treg cell formation and inhibiting IFN-γ production. A further periodontitis model in rats also proved its effectiveness in treating periodontitis [50,51,52]. In addition, studies have shown that TNF-α stimulation could change the contents of CEVs to enhance the anti-inflammatory effect. Nakao et al. found TNF-α pre-treatment could promote the secretion of CD73-rich GMSC-EVs to induce M2 macrophage polarization [53]. Similarly, chitosan hydrogels loaded with DPSC-derived CEVs (DPSC-EVs) reduced the expression of IL-23, IL-1α, TNF-α, IL-12, IL-1β, IL-27, and IL-17 in periodontitis tissues. By DPSC-EV-delivered *miR-1246*, *miR-125a-3p* and other miRNAs, macrophage phenotype was transformed from the pro-inflammatory M1 subtype to the anti-inflammatory M2 subtype [54,55].

On the other hand, ICEVs such as macrophage-derived CEVs (M-EVs) could also regulate M2 macrophage polarization and reduce inflammation through miRNAs such as *miR-99a*/*146b*/*378a* [44]. Moreover, M2 macrophage-derived CEVs (M2-EVs) changed the proportion of different types of macrophages and increase the M2/M1 ratio in bone defect models [26]. M2-EVs were reported to be abundant in IL-10, which could be delivered to target cells such as macrophages and BMMSCs to enhance their anti-inflammatory functions [56]. In addition, the role of Treg cell-derived CEVs (Treg-EVs) in immune tolerance and inflammation inhibition is also worth attention [57,58].

### 2.3. CEVs Promote Periodontal Bone Regeneration

Alveolar bone regeneration is the most studied part in periodontal regeneration. In terms of treatment, SCEVs and ICEVs promoted stem cell adhesion, migration, proliferation and osteogenic differentiation. The merits of CEVs in bone tissue repair and regeneration has been broadly confirmed in bone fracture, bone defect, implant bone integration, periodontal defect, periodontitis and other models [26,59,60,61,62,63,64].

It can be concluded that SCEVs mediate periodontal bone formation mainly via regulating periodontal cell function and fate, modulating immune and inflammatory microenvironment, inhibiting osteoclast formation, promoting vascularization, enhancing antibacterial effects, delivering active ingredients, and interfering with multiple signaling pathways. Relevant study showed that CD73 on the surface of mesenchymal stem cell (MSC)-derived CEVs (MSC-EVs) activated the Akt and ERK signaling pathways to promote PDLC migration and proliferation. The collagen sponge loaded with MSC-EVs facilitated new bone formation in the periodontal defect area of rats [63]. SHED-EVs were proved to promote BMMSCs proliferation, osteogenic differentiation and also angiogenesis through AMPK signaling pathway. Wnt3a and BMP2 in SHED-EVs activated Wnt/β-catenin and BMP/Smad signaling pathways, respectively, to promote the osteogenic differentiation of PDLSCs, thereby playing a protective role in periodontal bone homeostasis [46,65,66]. Mohammed et al. demonstrated that adipose-derived stem cells (ADSCs) and the CEVs derived from them (ADSC-EVs) could be applied as an adjunct therapy for periodontitis treatment in rats. Particularly, ADSC-EVs promoted PDL repair and osteoid tissue generation [62]. BMMSC-EVs promoted TGF-β1 expression, migration, proliferation, osteogenic differentiation of PDLCs. At the same time, the M2 polarization of macrophages was induced and the activity of osteoclasts was inhibited by BMMSC-EVs. The nanocomposite hydrogel integrated with BMMSC-EVs could effectively repair the bone resorption in periodontitis rats [67]. Additionally, dental pulp (stem) cell (DPC/DPSC)-derived CEVs (DPC/DPSC-EVs) could facilitate MC3T3-E1 migration, suppress osteoclast formation, transfer *miR-1246* to induce M2 macrophage polarization, and alleviate bone destruction in periodontitis mice [54,68]. In addition to regulating the phenotype of macrophages, SCEVs such as GMSC-EVs could inhibit T cell activation, increase the number of Treg cells, and finally promote bone remodeling in a rat periodontal defect model [50].

Increasing evidence suggests that SCEVs from mild inflammatory cytokine pre-stimulated cells are beneficial to periodontal bone formation. For example, CEVs derived from DPSCs pretreated with a low concentration of TNF-α (TNF-α-DPSC-EVs) delivered *miR-758-5p* targeting LMBR1 to activate BMP2/4 and promote the osteogenic differentiation of PDLSCs [69]. Similarly, LPS-pretreated dental follicle (stem) cell (DFC/DFSC)-derived CEVs (LPS-DFC-EVs/LPS-DFSC-EVs) promoted the migration, proliferation and osteogenic differentiation of PDLSCs, and exerted an antioxidant function through ROS/MAPK signaling pathway. By down-regulating RANKL/OPG ratio and reshaping macrophage phenotype, it accelerated bone reconstruction in periodontitis and periodontal defect models of rats and Beagle dogs [70,71,72]. Further, inflammatory induction, gene modification and mechanical stimulation also endowed SCEVs the effectiveness in promoting the osteogenic differentiation of stem cells [29,73,74].

Many studies demonstrated that ICEVs favored bone regeneration through remodeling the immune microenvironment and delivering bioactive molecules such as miRNAs. Several subtypes of macrophages present in various stages of bone healing, among which osteoclasts are the typical representatives for bone resorption, and osteomacs in bone remodeling units are the resident macrophages [75]. Macrophages, especially M2 macrophages, could promote periodontal and other bone regeneration [76,77,78,79]. To a large extent, M-EVs mediated the macrophage-based bone regeneration. It was found that both M-EVs and M2-EVs could assist skull defect repair in rats and femur fracture healing in mice, between which, M2-EVs were more effective [26,80]. It has also been reported that M2-EVs were inferior to M1 macrophage-derived CEVs (M1-EVs) in osteogenic induction [81]. However, most studies confirmed that M2-EVs could promote the osteogenesis of BMMSCs by delivering multiple miRNAs, such as *miR-690*, *miR-5106* and *miR-26a-5p* [80,82,83]. In terms of periodontal bone regeneration, Liao et al. pointed out that M2-EVs might also promote the osteogenic differentiation of PDLSCs through a variety of miRNAs [84]. In addition to miRNAs, Chen et al. found that M2-EVs were rich in IL-10, which promoted BMMSC osteogenic differentiation through IL-10/IL-10R signaling pathway, inhibited osteoclast formation, and reduced bone resorption in mice with periodontitis [56]. Liu et al. investigated the effect of materials on the osteoinductive activity of M-EVs. Intrafibrillarly mineralized collagen (IMC) was found to be superior than extrafibrillarly mineralized collagen (EMC) in promoting M2 macrophage formation and rat mandibular defect repairing. IMC-induced macrophage-derived CEVs (IMC-EVs) improved MSC osteogenic differentiation through BMP2/Smad5 signaling pathway [85]. In addition to the material itself, physical topography, BMP-2 and Mg2^+^ pre-stimulated macrophage-derived EVs all showed higher osteogenic induction capacity via delivering *miR-381* or activating autophagy [64,86,87].

Similarly, CEVs derived from monocytes (MC-EVs) induced the up-regulated expression of osteogenic-related genes in MSCs [88]. Schwann cell-derived CEVs (SC-EVs) promoted BMMSC osteogenic differentiation through TGF-β1/SMAD2/3 signaling pathway, M2 macrophage polarization, vascularization, nerve repair, and ultimately bone defect restoration [89]. Inflammatory osteoclast (iOC)-derived CEVs (iOC-EVs) have been proved to reduce the Osterix ubiquitination in MC3T3-E1 cells by transferring lncRNA LIOCE, thus facilitating bone regeneration [90]. In addition, Treg cells could play a positive role in bone regeneration by inhibiting over-activated T cells, reducing the proportion of Th1 and Th17 cells, and eliminating inflammation [77,91]. However, research on Treg-EVs in the field of regeneration is still lacking. In view of its role in remodeling M2 macrophage polarization and improving inflammatory microenvironment, the future of Treg-EV-based periodontal regeneration is promising [92,93].

### 2.4. CEVs Promote Cementogenesis

Cementum shares similarities with bone in structure, matrix composition, source of ectodermal mesenchymal progenitor cells. They also both secret non-collagenous proteins such as BSP, OPN, OCN, vitronectin and fibronectin [7,94]. So, many studies have looked at the two together [95,96,97,98]. From this point of view, SCEVs and ICEVs, which have positive effects on osteogenesis, can theoretically promote cementogenesis. However, different from periodontal bone regeneration, there is no typical cementum regeneration model due to the special anatomical position and structure of cementum and the lack of characteristic markers. Currently, periodontal defect model was the commonly used model to identify the newly formed cementum [99]. We previously used cementoblasts combined with collagen sponges to conduct subcutaneous ectopic transplantation in nude mice. In addition, other scholars also carried out studies on the establishment of root resorption model with cementum destruction under orthodontic force. However, none of the above models is mature [100,101]. Therefore, relevant studies mainly stay at the cellular level.

A variety of stem cells have been shown to be associated with cementogenesis, including Gli1^+^ PDL progenitor cells, PDL(S)Cs, DPSCs, periapical follicular stem cells (PAFSCs), and BMMSCs [95,98,102,103,104]. To our knowledge, the effects of SCEVs on cementum formation have not been explored yet. Whether SCEVs are beneficial to cementum formation as much as bone formation remains to be verified.

For immune cells, macrophages especially M2 macrophages are essential for cementum formation. Li et al. found that M2 macrophages promoted the cementoblastic differentiation of PDLSCs by secreting IL-10, VEGF and other cytokines to activate Akt signaling pathway [19]. Our recent work found that M2 macrophages possessed the chemotaxis to inflammatory microenvironment with high level of CCL2 from cementoblasts, and rescued *P. gingivalis*-suppressed cementoblast mineralization by activating p38 signaling pathway [105]. The effects of ICEVs on cementum formation was also poorly studied. Zhao et al. proved that M2-EVs could promote cementoblast mineralization. Additionally, they used the orthodontic tooth movement strategy combined with gingival crevicular injection of LPS to construct the cementum absorption model, followed by the injection of different kinds of M-EVs. Their results indicated that M2-EVs could also significantly promote cementoblast mineralization in vivo [106]. Our unpublished study demonstrated that Ckip-1 (casein kinase 2-interacting protein 1)-silenced macrophage-derived CEVs (sh-Ckip-1-EVs) possesses the characteristic of M2-EVs in facilitating cementoblast mineralization. It can be seen that cementum regeneration based on SCEVs and ICEVs still has great exploration space.

### 2.5. CEVs Promote Angiogenesis and Periodontal Soft Tissue Remodeling

Angiogenesis is essential for periodontal regeneration, which can synergically enhance the osteogenic differentiation of PDLSCs and other stem cells. The growth and remodeling of the skeletal system, including alveolar bone, are often accompanied by angiogenesis [107,108]. SCEVs are closely associated with angiogenesis. An abnormal increase in vascularization in gingiva and PDL due to inflammatory cell infiltration is among the most common features leading to the progression of periodontitis, and SCEVs may play a role in it. Research has revealed that down-regulation of *miR-17-5p* in inflammatory PDLSCs (iPDLSCs) leads to the up-regulation of VEGFA, which could be packed into iPDLSC-derived CEVs (iPDLSC-EVs), and then delivered to human umbilical vein endothelial cells (HUVECs) to enhance angiogenesis. Similar results were also verified in inflammatory DPSC (iDPSC)-derived CEVs (iDPSC-EVs) [109,110,111,112,113].

On the other hand, various SCEVs were found to promote angiogenesis, and further the bone regeneration by bioactive factors and improved HUVEC functions [114,115,116,117]. Pizzicannella et al. constructed 3D collagen membranes loaded with PDLSCs and polyethylenimine (PEI)-coated PDLSC-EVs (PEI-PDLSC-EVs). They found the membranes promoted bone defect repair by increasing the expression of VEGF, VEGFR2, and osteogenesis-related markers in PDLSCs [116]. One related study also showed that BMMSC-EVs promoted the proliferation, migration and tube formation capacity of HUVECs through HIF-1α/VEGF signaling pathway, thereby benefiting bone reconstruction [118,119]. Similar functions were demonstrated in umbilical cord mesenchymal stem cell (UCMSC)-derived CEVs (UCMSC-EVs), ADSC-EVs and DPSC-EVs [65,120,121,122]. CEV-based angiogenesis was also studied in periodontal regeneration. For example, SHED-EVs were found to facilitate HUVEC functions, and effectively promote the angiogenesis and bone regeneration in the periodontal defect areas of rats [65]. ADSC-EVs could be used to increase the number of small blood vessels in PDL and promote its’ remodeling [62].

The relationship between ICEVs and angiogenesis has also been studied. Aharon et al. found that in addition to regulating thrombosis and endothelial cell apoptosis, MC-EVs also significantly promoted HUVECs’ tube formation capacity [123]. Osteoclast-derived CEVs (OC-EVs) could improve the functions of HUVECs and CD31 expression. This angiogenesis-promoting effect could be enhanced ulteriorly by compression force [124]. The adverse effects of M-EVs on HUVEC function under certain conditions have ever been reported. However, M2-EVs could inhibit inflammation and apoptosis of HUVECs, and promote their functions by delivering *miR-221-3p* and *miR-21* activating related signaling pathways [89,125,126]. Therefore, the degree of vascularization of tissues or scaffolds should be taken into consideration in a successful periodontal regeneration strategy. The collaboration of CEVs to enhance periodontal blood vessel and bone regeneration is a potential entry direction for periodontal regeneration.

The process of CEV-based periodontal regeneration is more or less accompanied by the remodeling of periodontal soft tissues including gingiva and PDL. The relevant studies mainly focus on SCEVs. Huang et al. showed that DFC-EVs, especially LPS-DFC-EVs, could reduce the number of *P. gingivalis* in the gingival tissues of mice with periodontitis, and the expression of *IL-6* and *IL-1β* in gingiva and PDL [127]. Liu et al. constructed a mouse model of palatal gingival defect. Based on it, they found stem cell from apical papilla (SCAP)-derived CEVs (SCAP-EVs) promoted angiogenesis, type I collagen and fibronectin expression, gingival epithelium and connective tissue regeneration through Cdc42 [128]. PDL(S)Cs are essential cell types in PDL. The above mentioned MSC-EVs, BMMSC-EVs, ADSC-EVs, DPSC-EVs, DFSC-EVs and PDLC-EVs, which have therapeutic potential in periodontal bone regeneration, could not only promote the osteogenic/cementogenic differentiation of PDL(S)Cs, but also enhance their proliferation and migration ability. Further, they could improve blood and nutrient transport and alleviate inflammatory microenvironment, which ultimately promoting periodontal epithelium healing and PDL remodeling [47,54,62,63,67,70].

In summary, CEVs represented by SCEVs and ICEVs play an important role in the regeneration of periodontal hard and soft tissues. Compared to ICEVs, the researches on SCEVs are more mature. The above studies have emphasized the positive effects of CEVs on immune modulation, inflammation clearance, cell function regulation, angiogenesis, stem cell differentiation, periodontal tissue remodeling, etc., providing an effective strategy for CEV-based periodontal regeneration (Table 1).

## 3. BEVs and Periodontal Regeneration

### 3.1. Classification and Biogenesis of BEVs

Bacteria can release spherical membrane vesicles (20 to 400 nm) with lipid layers. These membrane vesicles affect different biological processes, including virulence, horizontal gene transfer, export of cellular metabolites, phage infection, and intercellular communications. BEVs can be classified into outer membrane vesicles (OMVs), outer-inner membrane vesicles (OIMVs), cytoplasmic membrane vesicles (CMVs), and tube-shaped membranous structures (TSMSs) [129].

With no outer membrane, Gram-positive bacteria mainly secrete CMVs, which are formed by endolysin-triggered ‘bubbling cell death’. CMVs contain cytosolic components and membrane components. In Gram-negative bacteria, budding from the outer membrane gives rise to formation of OMVs, which do not contain cytoplasmic components. Explosive cell lysis is caused by phage-produced endotoxin, which degrades the peptidoglycan cell wall to produce explosive outer membrane vesicles (EOMVs) and OIMVs. OIMVs can specifically encapsulate DNA, and EOMVs randomly contain cytoplasmic components compared to blister-forming OMVs [130,131]. Many factors can influence the biogenesis of OMVs. (1) The production of OMVs increases in regions with reduced Braun’s lipoprotein–peptidoglycan (Lpp-PG) crosslinks. (2) Insertion of the Pseudomonas quinolone signal (PQS) into the outer leaflet of the outer membrane can increase membrane curvature, leading to OMVs formation. (3) Increased OMVs production can be affected by accumulation of misfolded proteins or envelope components (such as lipopolysaccharide or peptidoglycan fragments), which lead to cross-linking occurring and outer membrane bulging. (4) Some lipid microdomains of the outer membrane have the tendency to bulge outward due to their charge, and this bulging leads to increased generation of OMVs [132].

Among the above-mentioned BEVs, studies related to OMVs are the most abundant and mature. In recent years, OMVs have received more and more attention in pathogenic mechanism and nanomaterial application. In the following, we mainly summarize the pathogenicity of OMVs, and their applications and prospects in periodontal regeneration.

### 3.2. Pathogenicity of BEVs in Periodontal Tissues

Firstly, as a typical representative of BEVs, OMVs have a protective effect on the bacteria themselves. Bacteria can respond to threats from the external environment by regulating OMVs production and components. For example, OMVs could protect bacteria by expelling misfolded proteins or toxic substances from the periplasmic cavity [133]. Secondly, OMVs play an important role in bacterial nutrient uptake; DNA and proteins in OMVs could serve as a direct source of nitrogen and phosphorus for bacterial growth. Specially, they could also transport hydrolases and polysaccharide lytic enzymes that assist bacteria lacking these enzymes to metabolize polysaccharides for nutrients [134,135]. Thirdly, OMVs can act as pseudo-targets for some antibiotics, protecting bacteria by absorbing or binding antibiotics. OMVs of *Acinetobacter baylyi* could enable negative strains to acquire resistance genes through horizontal gene transfer [136]. Moreover, OMVs can deliver virulence factors into the host, compromising the host immune system and thus promoting bacterial infection [137]. In addition, the threat of OMVs to the host cannot be ignored. OMVs are closely linked to epithelial cells and innate immune cells. Epithelial cells are the first line of defense of the host. On the one hand, OMVs could be recognized by pattern recognition receptors on the cell surface and induce or inhibit the production of cytokines and chemokines by epithelial cells. On the other hand, it could disrupt the tight junctions between epithelial cells and mucosal integrity, allowing the virulence factors to enter the submucosa and contact with neutrophils, dendritic cells, and macrophages [138]. Finally, after entering into the host cells, OMVs were recognized and then degraded, causing an inflammatory response, activating dendritic cells and promoting T-cell immunity. Activated CD40 ligands on the surface of helper T cells bind to homologous receptors on the surface of B cells, facilitating B cell proliferation and differentiation into effector B cells and memory B cells, facilitating antibody production [139,140,141].

There is a complex microbial ecosystem exists in the oral cavity of periodontitis patients, with the main pathogenic bacteria “red complex” (*P. gingivalis*, *T. forsythia* and *T. denticola*) [142]. Other Gram-negative bacteria such as *A. actinomycetemcomitans* and *F. nucleatum* also play an essential role in the development and progression of periodontitis [143]. The pathogenicity of BEVs in periodontal disease has been extensively studied. Researchers performed genomic DNA analysis of salivary vesicles from healthy individuals and patients with periodontal diseases. It was found that *P. gingivalis*-derived OMVs (*Pg*-EVs) were significantly increased in periodontitis patients [144]. Mass spectrometry-based high-throughput proteomic analysis revealed that gingipain, leukotoxin and peptidylarginine deiminase were enriched in OMVs from periodontal pathogenic bacteria [145]. Further, small RNAs could be delivered to the host cellular immune system via OMVs of *P. gingivalis*, *A. actinomycetemcomitans*, and *T. forsythia* [146]. For example, *Pg*-EVs can target chromobox 5 (CBX5) of PDLCs through *sRNA45033* in OMVs, and promote apoptosis by regulating the methylation of p53 [147]. OMVs of periodontal pathogenic bacteria can also interact with host immune cells. OMVs from the “red complex” induced maturation of bone marrow dendritic cells and differentiation of naive CD4^+^ T cells into Th1 or Th17 cells [148]. *F. nucleatum*-derived OMVs (*Fn*-EVs) tended to polarize M0 macrophages to M1 macrophages, thereby exacerbating the alveolar bone destruction caused by periodontitis [149]. OMVs of periodontal pathogenic bacteria also have an effect on other periodontal pathogenic bacteria. For example, *Pg*-EVs could down-regulate the expression of the surface adhesion proteins FadA and FomA of *F. nucleatum*, and thus inhibiting the invasion ability of *F. nucleatum* [150]. The related literature further illustrates that BEVs can disrupt periodontal homeostasis in a variety of ways, including through virulence factors, promoting plaque biofilm formation, invading the immune system, activating inflammatory responses, damaging periodontal tissues, and inducing systemic diseases [29].

### 3.3. Potential Therapeutic Values of BEVs in Periodontal Treatment and Regeneration

As mentioned above, the pathological roles and mechanisms of BEVs in periodontitis have been extensively studied. However, their research in periodontal treatment and regeneration is still very scarce. According to previous studies, their potential value in periodontal treatment mainly includes the following aspects.

BEVs can be used as a vaccine for periodontitis. Some researchers have proposed that *Pg*-EVs could be stored for a long time for its structurally and functionally stability and resistant to proteinase K. Most importantly, it could carry *Pg*-EVs components into the host immune system. Mice immunized intranasally with OMVs + TLR3 agonist (Poly(I:C)) showed a significant reduction in the amount of *P. gingivalis* in the oral cavity, and elevated levels of serum IgG (including IgG1 and IgG2a) and s-IgA in saliva. The safety of *Pg*-EVs was also confirmed by intracerebral injection experiments [151]. Similar studies further suggested that *Pg*-EVs possessed strong immunogenicity and antigenicity, which may originate from the LPS and A-LPS-modified proteins in OMVs [152]. Furthermore, other research also confirmed the potential of other OMVs in vaccine development [153]. Therefore, *Pg*-EVs or other OMVs of periodontal pathogenic origin are likely to be an effective vaccine strategy for periodontitis in terms of bacterial clearance, safety, low cost, high efficiency, and ease of transport [154]. Overall, a reduction in *P. gingivalis*, a key pathogenic bacterium in periodontitis, would be beneficial in reducing the destruction of periodontal tissues and enhancing periodontal regeneration.

BEVs can act as drug delivery vehicles to exert antibacterial effects. periodontal pathogenic bacteria such as *P. gingivalis* and *A. actinomycetemcomitans* could evade antibiotics by parasitizing human epithelial cells. High permeability of BEVs gave the possibility for drugs such as antibiotics to enter the cells and kill the bacteria efficiently [155]. Huang et al. found that the application of sublethal doses of quinolone antibiotics to bacterial culture media induced bacteria to load high doses of antibiotics in OMVs. There was an increase in the number of particles, protein content and particle size of the vesicles after antibiotic treatment, while the nanostructure of the vesicles was not affected. Such highly biocompatible antibiotic-loaded vesicles could efficiently invade bacteria and release drugs, thereby effectively killing a wide range of Gram-negative bacteria and significantly reducing the amount of bacteria in the small intestine and feces of infected mice [156]. Thus, we hypothesize that drug-loaded BEVs could likewise be an effective strategy for suppressing periodontal pathogenic bacteria. However, their effectiveness in periodontal regeneration needs to be further investigated.

Saliva-derived EVs containing BEVs can be used for early detection of periodontitis. Han et al. showed that BEVs could circulate in saliva. Saliva-derived EVs were more suitable for biomarkers for early detection of periodontal status than saliva itself. BEVs can reflect the changes in periodontal bacteria. Researchers examined saliva-derived EVs and found a high percentage of periodontal pathogenic bacteria (*T. denticola*, *E. corrodens*, *P. gingivalis*, and *F. nucleatum*)-related BEVs in periodontitis patients [144]. For the benefits of early detection and prevention of periodontitis, BEVs could be used for the assessment of oral microecological environment in periodontal regeneration state.

Lastly, BEVs act as biological shuttle systems to detect molecular transfer between microorganisms and host cells. Miriam Bittel et al. applied *E. coli* to express Cre recombinase and produced OMVs loaded with Cre recombinase. Then, Cre protein activated the Cre-LoxP reporter system of genetically engineered murine intestinal cells and express tdTomato fluorescent protein. This technique could enable biomolecular visualization of biomolecular transferring. In addition to the intestine, bacterial-derived Cre induced extended marker gene expression in a wide range of host tissues, including heart, liver, spleen, kidney, and brain [157]. By analogy, periodontal pathogenic bacteria-derived OMVs expressing Cre recombinase could theoretically be used to visualize periodontal-systemic disease correlations, and even facilitate the development of visualized drug delivery-based periodontal therapy and regeneration. However, compared to CEVs, prospects for translation to therapy are far off for the reason that the role of BEVs in periodontal regeneration is limited and indirect. Thus, further applications need to be explored.

## 4. PEVs and Periodontal Regeneration

### 4.1. Classification and Biogenesis of PEVs

In this review, PEVs refers to the general term for vesicular nanoparticles isolated from plants. They have also been named as plant-derived exosome-like nanoparticles (PELNs), plant-derived vesicle-like nanoparticles (PDVLNs), plant-derived nanovesicles (PDNVs), etc., including the vesicles from extracellular apoplastic fluids and from tissue crushing fluids [158]. PEVs and mammalian EVs are similar in terms of morphology, size and other physical characteristics. PEVs are also cup-shaped with different particle sizes in different plants. Even in the same plant, there are vesicles with various sizes ranging from 50 to 500 nm. Compared with mammalian EVs, the study of PEVs is still in their infancy. Due to the unstandardized means of isolation and physical characterization of PEVs, and the lack of generic or specific markers of PEVs and plant species antibodies, it is difficult to use the nomenclature for CEVs to similarly classify PEVs [32,158,159].

PEVs can be secreted into the apoplast space. Compared with CEVs, the biological mechanism of PEVs formation is less studied, containing the following secretion models. (1) MVBs is a conventional pathway similar to exosome secretion. Initially, Halperin and Jensen observed the MVBs fused with the plasma membrane of carrot cells, followed by the release of vesicle-like structures into the wall spaces. Similar phenomenon was also observed in Arabidopsis infected with turnip mosaic virus [160,161]. (2) Exocyst positive organelles (EXPOs) pathway is a plant-specific secretory form. EXPOs are spherical, bilayer structures similar to autophagosomes. It has been reported that the outer membrane of EXPOs in Arabidopsis and tobacco cells could fuse with the plasma membrane to release monolayer vesicles into the apoplast space. EXPOs were also found in other cell types, such as root hair cells, root tips, and pollen grains [162]. (3) Vacuole, the dynamic organelles of plant cells, is another unconventional secretion approach. One related study confirmed that vacuoles of Arabidopsis cells, could mediate the auto-immune process by fusing with the plasma membrane to release antibacterial and cell death-inducing proteins [163]. In addition, vacuole pathway and MVBs pathway are closely related and intersected [164]. (4) In addition, the autophagosome pathway has also been reported to potentially present in plants [165].

PEVs are rich in sources and can be isolated from the fruits, roots, stems and leaves of various plants such as ginseng, sunflower seeds, oats, Arabidopsis, ginger, broccoli, bitter melon, grape, grapefruit, lemon, blueberry and orange. Small nucleic acids, proteins, lipids and small-molecule metabolites contained in PEVs are the main functional substances [166]. PEVs have fewer ethical problems in clinical transformation due to their plant origin, with advantages of low immunogenicity and wide availability. Hence, they can be used as safe and economical therapeutic drugs and delivery carriers [167,168]. In recent years, increasing evidence suggests that PEVs possess great potential to modulate immunity, inflammation, microbiome, and tissue regeneration [169]. Here, we mainly focus on their functions to summarize and prospect the therapeutic effects of PEVs in periodontitis treatment and periodontal regeneration.

### 4.2. PEVs Regulate Immune and Inflammatory Responses

PEVs have been demonstrated to regulate inflammation, antioxidant factor expression, and macrophage polarization. One recent study showed that orally administrated ginger-derived PEVs (Gr-EVs) could be absorbed by mouse intestinal macrophages, then promoted the expression of HO-1, IL-10 and Nrf2, and activated the anti-inflammatory Wnt/Tcf4 signaling pathway [170]. In addition to up-regulating the expression of IL-10 and HO-1, tea leaf-derived PEVs (TL-EVs) also decreased the expression of pro-inflammatory cytokines TNF-α, IL-6, IL-12, and ROS in LPS-stimulated macrophages [171]. Gin-EVs and shiitake mushroom-derived PEVs (SM-EVs) were involved in the inhibition of NLRP3 inflammasome assembly, IL-1β and IL-18 secretion, and pyroptosis in macrophages [172,173]. Cabbage-derived PEVs (Ce-EVs) and red cabbage-derived PEVs (RCe-EVs) were reported to inhibit the M1 macrophage polarization, and the apoptosis of keratinocytes and fibroblasts [174]. Similar anti-inflammatory, antioxidant and macrophage-regulatory effects have also been confirmed in PEVs derived from Aloe saponaria (AS-EVs), aloe vera peels (AVP-EVs), and grapefruits (GF-EVs) [175,176,177].

PEVs could regulate immune and inflammatory responses through a variety of components and signaling pathways. (1) miRNAs. Aquilano et al. showed that PEVs derived from dried nuts (Nut-EVs) could deliver *miR-159a* and *miR-156c* to reduce the expression of Tnfrsf1a protein in adipocytes and inhibit the TNF-α signaling pathway, to exert anti-inflammatory effects [178]. (2) Lipids. Deng et al. reported that Broccoli-derived PEVs (Bi-EVs) activated the AMPK signaling via lipid components, inhibited dendritic cell activation, induced tolerant dendritic cell formation, and ultimately alleviated DSS-induced colitis in mice [179]. (3) Proteins. Sriwastva et al. proved that Mulberry bark derived PEVs (MB-EVs) could alleviate mice colitis by activating AhR/COPS8 signaling pathway through HSPA8 [180]. (4) Phenolic compounds. Paiotti et al. also reported that grape juice containing phenolic compounds was protective against TNBS-induced colitis [181]. The same results have been obtained in various inflammatory models such as colitis and viral pneumonia [182,183]. However, PEVs were poorly studied in periodontal inflammation with only one related study. In this study, phosphatidic acid in Gr-EVs was demonstrated to reduce *P. gingivalis*-induced periodontal inflammation, characterized by decreased expression of *IL-1β*, *IL-6*, *IL-8* and *TNF-α* in periodontal tissues, and reduced inflammatory infiltration in PDL [184]. All the above studies revealed the important role of PEVs in regulating immune and inflammatory responses, which are also the manifestations of periodontitis. Their potential application values in the treatment of periodontitis and periodontal regeneration need to be further explored.

### 4.3. PEVs Reshape Microecology

On the other hand, periodontitis is a secondary inflammatory reaction caused by the imbalance of oral microecology. Maintaining the dynamic balance of oral microecology is vital to control the progression of periodontitis and improve the regeneration [1,2]. PEVs can mediate the interaction between plants and microorganisms. Under the stimulation of pathogens, plants themselves can secrete PEVs carrying a variety of sRNA and lipid components, playing an important role in the auto-immune process against bacteria and fungi. In addition, the mutualism between plants and microorganisms also involves the participation of PEVs [158,185]. In terms of therapy, PEVs could enhance the growth of probiotics. Lei et al. reported that lemon-derived PEVs (Ln-EVs) inhibited the production of Msp1 and Msp3 through RNaseP-mediated degradation of specific tRNA, thereby improving the bile tolerance of Lactobacillus rhamnosus GG [186]. PEVs could also inhibit the keystone periodontal pathogen. Sundaram et al. showed that the phosphatidic acid (34:2) in Gr-EVs interacted with the HBP35 on the outer membrane of *P. gingivalis* to mediate the internalization of Gr-EVs. The internalized Gr-EVs further inhibited the proliferation, growth, adhesion and invasion of the bacteria and reducing its virulence [184]. Therefore, regulating microecology through PEVs may potentially be an effective approach for periodontitis treatment and periodontal regeneration.

### 4.4. The Therapeutic Values of PEVs in Periodontal Regeneration

CEVs showed positive roles in various tissue regeneration. Given the similarity between PEVs and CEVs in structure and composition, PEVs also have potential values in tissue regeneration theoretically [187]. PEVs can enhance the proliferation and migration of various soft tissue-related cells, as well as angiogenesis, and have therapeutic effects on tissue healing and repair. AS-EVs were proved to enhance the proliferation and migration of human dermal fibroblasts (HDFs), and promote tube formation capacity of HUVECs [175]. In addition to improving HUVEC functions, GF-EVs were demonstrated to promote the proliferation, migration, and the expression of soft tissue healing-related markers such as type I collagen, fibronectin, laminin and vimentin in human immortalized epidermal cells (HaCaT cells) [176]. Similarly, cell function-promoting effects of wheat-derived PEVs (Wt-EVs) was also shown in HUVECs, HDFs and HaCaT cells [188]. In summary, these studies all revealed the potential role of PEVs in periodontal angiogenesis and soft tissue regeneration.

PEVs also regulate the function and fate of stem cells by their components. Ju et al. showed that grape-derived PEVs (Ge-EVs) contribute to the expression of growth-associated genes of Lgr5^+^ intestinal stem cells and their proliferation by activating the Wnt/β-catenin signaling pathway through their lipid components, thereby favoring the repair of the intestinal epithelium stimulated with DSS [183]. Strawberry-derived PEVs (Sy-EVs) promoted the early proliferation of ADSCs, and enhanced their antioxidant ability to reduce ROS production through vitamin C [189]. Ginseng-derived PEVs (Gg-EVs) could promote the neural differentiation of BMMSCs in vivo and in vitro by delivering miRNAs [190]. Therefore, it is reasonable to speculate that PEVs may also have positive regulatory effects on the proliferation, differentiation and other functions of various periodontal stem cells. Relevant works need to be further expanded.

PEVs are beneficial to periodontal hard tissue repair and regeneration. A recent study demonstrated that yam-derived PEVs (Yam-EVs) improved osteoblast mineralization for bone regeneration in ovariectomized (OVX)-induced osteoporotic mice by activating the BMP-2/p-p38-dependent Runx2 pathway [191]. As mentioned above, Sundaram et al. confirmed that Gr-EVs influenced a series of functions of *P. gingivalis* and reduced the periodontal inflammation through phosphatidic acid. They could also further reduce the number of Trap-positive osteoclasts, increase the number of Runx2-positive osteoblasts, and relieve the alveolar bone loss induced by *P. gingivalis* [184].

In conclusion, research on PEV-based periodontal regeneration is still lacking. Similar to CEVs, PEVs can also carry a variety of effective components to target cells to be involved in soft and hard tissue remodeling, which were expected to become a new strategy for periodontal regeneration.

## 5. Conclusions, Challenges and Prospects

EVs can be secreted by almost all cells in mammals, bacteria and plants, and the production of EVs runs through the whole life process. It seems that it is not a random event, but an active behavior of living organisms trying to change their surroundings. For example, CEVs mediate the interaction between eukaryotic cells, BEVs attack host cells, and PEVs resist the invasion of exogenous pathogens [192]. It is a great progress in medical field to be familiar with, understand and then master the characteristics and rules of EVs, and apply it in disease treatment and regenerative therapy. Periodontitis is an immune- inflammatory response caused by periodontal pathogens, involving immunity, inflammation, periodontal microecology, periodontal tissue destruction and many other aspects. EV-based periodontal regeneration is a new and promising strategy for repairing and rebuilding damaged periodontal tissues.

At present, there also exist some common issues to be improved in EV-based periodontal therapy, such as standardizing EV extraction methods, exploring new combined methods that can improve EVs yield and purity, reducing the clearance of EVs and increasing its targeting ability in vivo, and developing new multifunctional biomaterials with EVs retention capacity [29]. CEVs, BEVs and PEVs have similar physical properties such as morphology, structure, size and electric potential. They all have heterogeneity in many aspects and can carry different types of substances such as small RNAs, proteins, lipids and metabolites to target cells to perform specific functions. There are also differences. These include the functional substances, the amount and cost of access, and the possibility of clinical transformation. Compared with CEVs and BEVs, PEVs need further research in terms of extraction method and naming standardization, subtype classification, and marker discovery. In addition, research on BEVs mainly focus on the pathogenic effect in periodontitis, while CEVs and PEVs are more likely to be applied in periodontal regeneration. CEVs are maturely studied in periodontal treatment and regeneration, while PEVs are still in their infancy [29,167].

All in all, this review focuses on the multiple functions of EVs, such as immune regulation, anti-inflammation, reshaping microecology, regulating cell function and fate, and promoting soft and hard tissue regeneration. The latest research progress and potential application values of CEVs, BEVs and PEVs in periodontal regeneration are comprehensively expounded and discussed (Figure 1). In addition to CEVs, BEVs, and PEVs, some other EVs subtypes such as apoptotic cell-derived EVs (ApoEVs) and matrix vesicles (MVs) were also demonstrated to play positive roles in promoting osteogenesis, mineralization, and periodontal regeneration [193,194]. Moreover, the specific mechanisms of EV biogenesis, the development of EV-friendly scaffold materials, the synergistic function of EVs as a carrier to deliver genes and drugs, the transformation of waste into treasure (EVs in feces and urine), and the applications of engineered EVs are also the directions for further research in the field of periodontal regeneration. Although EV-based periodontal regeneration has made great progresses, we still face many unsolved problems, and its clinical transformation will be a long-term challenge. Nevertheless, it is only a matter of time. We should believe that it will have a bright future.

## Figures and Tables

**Figure 1 ijms-24-05790-f001:**
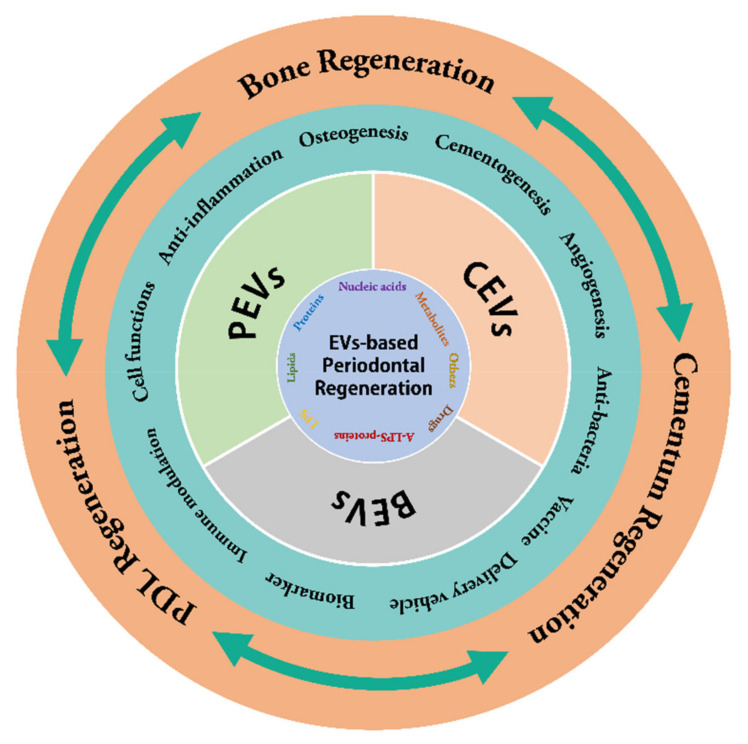
CEVs, BEVs and PEVs with versatile functions for periodontal regeneration. CEVs, eukaryocyte-derived extracellular vesicles; BEVs, bacterial-derived extracellular vesicles; PEVs, plant-derived extracellular vesicles.

**Table 1 ijms-24-05790-t001:** Application of SCEVs and ICEVs in periodontal regeneration.

Type	Regeneration Functions	Mechanisms	Scaffold/Amount of EVs	Location/Sacrifice Time	Ref.
MSC-EVs	Improve proliferation and migration of PDLCs, promote new bone formation in rat periodontal defects	Activate the Akt, ERK pathways via CD73	collagen sponge; 40 µg	Mesial side of the upper 1st molar; 2 w, 4 w	[63]
ADSC-EVs	Promote PDL repair and osteoid tissue formation in rats with periodontitis	/	PBS suspension; 80–150 µg	The lower incisors; 2 d, 2 w, 4 w	[62]
BMMSC-EVs	Improve proliferation, migration, osteogenic differentiation of PDLCs, promote bone regeneration in rats with periodontitis	Promote M2 macrophage polarization and TGF-β1 expression	Nanocomposite hydrogel; 1 time/week	Right upper 2nd molar; 4 w, 8 w	[67]
SHED-EVs	Improve proliferation, osteogenesis of BMMSCs, promote PDL remodeling and bone regeneration in mice with periodontitis	Reduce inflammation	PBS suspension; 20 µg/week	The upper 1st molar; 2 w	[46]
SHED-EVs	Improve the functions of HUVECs, promote osteogenesis of BMMSCs, promote angiogenesis and bone formation in rat periodontal defects	Activate the AMPK pathway	β-tricalcium phosphate; 100 µg	Buccal side from the lower 1st molar to 3rd molar; 4 w	[65]
DFSC-EVs	Improve proliferation, migration, osteogenic differentiation of PDLSCs, promote PDL remodeling and bone regeneration in rat periodontal defects	Activate p38 pathway	collagen sponge; /	Buccal side of the lower 1st molar; 2 w, 4 w, 8 w	[70]
LPS-DFC-EVs	Improve proliferation, migration, osteogenic differentiation of PDLSCs, promote PDL remodeling and bone regeneration in periodontitis rats	Decrease the RANKL/OPG ratio	Nanocomposite hydrogel; 50 µg/week	Right upper 2nd molar; 4 w, 8 w	[71]
LPS-DFSC-EVs	Improve proliferation, migration of PDLSCs, promote PDL remodeling and new bone formation in beagle dogs with periodontitis	Activate the ROS/MAPK pathway, promote M2 macrophage polarization	Hyaluronic acid hydrogel; 200 µg/week	The lower 3rd premolar; 8 w	[72]
GMSC-EVs	Promote bone regeneration in rat periodontal defects	Inhibit T cell activation, increase Treg cell formation	Polymer particle; ≈230 μg	Mesiolingual side of the upper 1st molar; 8 w	[50]
HHH-DPC-EVs	Improve migration of MC3T3-E1 cells, alleviate bone resorption in mice with periodontitis	Inhibit osteoclast formation	Saline solution; 7.5 × 10^8^/time, day 0, 2, 4	Right upper 2nd molar; 1 w	[68]
DPSC-EVs	Promote periodontal epithelial healing and bone regeneration in mice with periodontitis	Deliver *miR-1246* to promote M2 macrophage polarization	Chitosan hydrogel; 50 μg	Left upper 2nd molar; 4 w	[54]
M2-EVs	Improve osteogenesis of BMMSCs, alleviate bone resorption in mice with periodontitis	Deliver IL-10 to activate the IL-10/IL-10R pathway	PBS suspension; 3 µg/time, day 1, 4, 7	Right upper 2nd molar; 2 w	[56]

## Data Availability

Not applicable.

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
