# Peer review of "The Applications and Potentials of Extracellular Vesicles from Different Cell Sources in Periodontal Regeneration"

_ijms, 2023, doi:10.3390/ijms24065790_

Round 1

Reviewer 1 Report

The submitted manuscript is generally well organized and informative.

1. Title: The word 'different species' seems to mean different kinds of animals or kinds of plants. So consider changing to 'different cell types' or 'different cell sources'

2. page 5, line 210: References are incorrectly marked in the format or this journal. So please revise (Deng et al. 2022; Lv et al. 2020; Xu et al. 2020)

Reviewer 2 Report

Many thanks for the paper submission. This is a nicely written submission. However some modifications are required in order to proceed to publication.

This paper reports The applications and potentials of extracellular vesicles from 2 different species in periodontal regeneration.

1) at line 35 the authors should underline recent discoveries of periodontal disease, please add the following phrase:

"..Recently periodontal disease was reported as risk factor for infection after mandibular fracture and associated with low cognitive performance amongst older adults.

please cite the following

Marruganti, , Aimetti, Grandini, Sanz,  Romandini, . (2023). Periodontitis and low cognitive performance: A population‐based study. Journal of Clinical Periodontology. 10.1111/jcpe.13779.

Chisci G, Gabriele G, Gennaro P. Periodontal disease before and after fractures of the mandible. Br J Oral Maxillofac Surg. 2023 Jan;61(1):116. doi: 10.1016/j.bjoms.2022.09.016. Epub 2022 Nov 22.

2) at line 52 please modify the sentence into the following

"Research in periodontal regeneration included including guided tissue regeneration, platelet-rich plasma and fibrin, growth and differentiation factors, enamel matrix protein, and autologous graft, allograft and xenografts."

please cite the following

Barone A, Toti P, Quaranta A, Alfonsi F, Cucchi A, Calvo-Guirado JL, Negri B, Di Felice R, Covani U. Volumetric analysis of remodelling pattern after ridge preservation comparing use of two types of xenografts. A multicentre randomized clinical trial. Clin Oral Implants Res. 2016 Nov;27(11):e105-e115. doi: 10.1111/clr.12572.

Sassano P, Gennaro P, Chisci G, Gabriele G, Aboh IV, Mitro V, di Curzio P. Calvarial onlay graft and submental incision in treatment of atrophic edentulous mandibles: an approach to reduce postoperative complications. J Craniofac Surg. 2014;25(2):693-7. doi: 10.1097/SCS.0000000000000611.

Majzoub J, Ravida A, Starch-Jensen T, Tattan M, Suárez-López Del Amo F. The Influence of Different Grafting Materials on Alveolar Ridge Preservation: a Systematic Review. J Oral Maxillofac Res. 2019 Sep 5;10(3):e6. doi: 10.5037/jomr.2019.10306.

Chisci G, Fredianelli L. Therapeutic Efficacy of Bromelain in Alveolar ridge Preservation. Antibiotics (Basel). 2022 Nov 3;11(11):1542. doi: 10.3390/antibiotics11111542.

3) figure 1 should explain the abbreviations of CEVs, BEVs and PEV

   4) please report the keyword in alphabetical order

Round 2

Reviewer 2 Report

accept